# Gut Mucosal Proteins and Bacteriome Are Shaped by the Saturation Index of Dietary Lipids

**DOI:** 10.3390/nu11020418

**Published:** 2019-02-16

**Authors:** Nijiati Abulizi, Candice Quin, Kirsty Brown, Yee Kwan Chan, Sandeep K. Gill, Deanna L. Gibson

**Affiliations:** 1Department of Biology, IKBSAS, University of British Columbia, Okanagan campus, Kelowna V1V 1V7, Canada; nijiati.abulizi@gmail.com (N.A.); candicequin@hotmail.com (C.Q.); kirsty.brown12@gmail.com (K.B.); cyk.carol@hotmail.com (Y.K.C.); sand.gill01@gmail.com (S.K.G.); 2Department of Medicine, Faculty of Medicine, University of British Columbia, Vancouver V6T 1Z3, Canada

**Keywords:** Host-microbe interactions, gut microbiome, dietary lipids, polyunsaturated fatty acids, monounsaturated fatty acids, saturated fatty acids, proteome, 16S rRNA gene amplicon sequencing, short-chain fatty acid metabolism

## Abstract

The dynamics of the tripartite relationship between the host, gut bacteria and diet in the gut is relatively unknown. An imbalance between harmful and protective gut bacteria, termed dysbiosis, has been linked to many diseases and has most often been attributed to high-fat dietary intake. However, we recently clarified that the type of fat, not calories, were important in the development of murine colitis. To further understand the host-microbe dynamic in response to dietary lipids, we fed mice isocaloric high-fat diets containing either milk fat, corn oil or olive oil and performed 16S rRNA gene sequencing of the colon microbiome and mass spectrometry-based relative quantification of the colonic metaproteome. The corn oil diet, rich in omega-6 polyunsaturated fatty acids, increased the potential for pathobiont survival and invasion in an inflamed, oxidized and damaged gut while saturated fatty acids promoted compensatory inflammatory responses involved in tissue healing. We conclude that various lipids uniquely alter the host-microbe interaction in the gut. While high-fat consumption has a distinct impact on the gut microbiota, the type of fatty acids alters the relative microbial abundances and predicted functions. These results support that the type of fat are key to understanding the biological effects of high-fat diets on gut health.

## 1. Introduction

The mammalian gut has co-evolved with thousands of bacterial species and together they form a complex dynamic relationship of which the physiological consequences are largely undiscovered. The gut microbial ecosystem has the potential to influence the overall health status of the mammalian host by forming an interface between the gut mucosal surface and the luminal food ingested into the body. Molecular crosstalk between the microbiome and the host epithelium influences intestinal barrier function, in part through the release of microbial metabolites like short-chain fatty acids (SCFA) [1]. Increased intestinal permeability caused by a disruption of the microbiome, termed dysbiosis, has been implicated in diseases including inflammatory bowel disease (IBD), obesity and diabetes [2]. High-fat diets have been shown to induce dysbiosis, primarily characterized by the escalation of Firmicutes accompanied by a decrease of Bacteroidetes [3,4]. Changes in microbes induced through diet modulate major gene networks including signal transduction, inflammation, histamine, cell migration and adhesion [5]. Therefore, identifying specific nutrients that prevent dysbiosis may be important in preventing associated diseases.

Lipids are essential for normal development and survival in mammals. Subtle differences in the chemistry of fatty acids effect mammalian physiology and inflammation. Oleic acid, a monounsaturated fatty acid (MUFA), is the main component of olive oil, a major ingredient in the Mediterranean diet. In general, MUFA consumption is associated with health benefits including lower prevalence of digestive system cancers [6], decreased type 2 diabetes [7] and IBD [8]. Indeed, we have shown that olive oil diets are effective at protecting against murine colitis [9]. In contrast, while North American dietary guidelines recommend consuming omega-6 polyunsaturated fatty acids (n-6 PUFA), common in vegetable seed oils, excessive consumption of n-6 PUFA is a risk factor for IBD in humans [10]. In support of these findings, we have shown n-6 PUFA exacerbates murine colitis [9,11,12]. Conflicting data exists for dietary intake of saturated fatty acids (SFA), which have no double bonds and are found in dairy as well as coconut oil. SFA have been criticized as adversely affecting health over the past few decades, yet chronic inflammatory diseases are increasing while the global consumption of SFA have been in line with recommended low intakes [13]. Recently, a European prospective cohort study found that milk consumption is associated with decreased risk for IBD patients [14]. In contrast, SFA fed to mice resulted in increased spontaneous colitis in IL-10^-/-^ mice via conjugation of hepatic bile acids which promoted growth of *Bilophila wadsworthia* [15]. Yet, components of animal fat, such as butyric acid, suppress inflammation [16], protect against DSS-colitis [17] and stimulate colonic repair [18]. In line with this, we have shown that milk fat promotes beneficial responses during colitis [9]. While there is evidence that different dietary fatty acids have differential effects on host health, their effects on the gut bacterial ecosystem and their functional interaction with the host are not well explored.

To understand the tripartite relationship between lipid diet, gut bacteria and the host, we fed mice a 40% (by energy) isocaloric and isonitrogenous diet composed of either corn oil, olive oil or milk fat for 5 weeks post-weaning. The gut tissues were collected for 16S rRNA gene amplicon sequencing and metaproteomic analysis. We show the corn oil diet, rich in n-6 PUFA, produces a microbiome predicted to have enhanced virulence and pathogenicity potential. This was associated with a colonic proteome increased in proteins involved in inflammation, oxidative stress and barrier dysfunction. While the milk fat diet, rich in SFA, resulted in a host-microbe relationship indicative of inflammation, there was also a compensatory protective response evident by the increased host sirtuin signaling pathway and microbial production of SCFA. In marked contrast to both corn oil and milk fat, the olive oil diet, rich in MUFA resulted in a microbiome most similar to a low-fat diet. These results support that not all high-fat diets promote similar host and microbial responses and that consideration of the type of fat in high-fat diets is essential when investigating gut health. These results have the potential to guide evidence-based nutrition recommendations for IBD patients who can suffer from nutrient deficiencies from overly restrictive dietary regimes including low-fat diets.

## 2. Materials and Methods

### 2.1. Dietary Interventions and Tissue Collection

Three-week-old male and female C57BL/6 mice (total n=32, n=8 each diet; 4 each sex) were fed irradiated isocaloric, isonitrogenous diets for 5 weeks. High-fat diets contained 40% energy from olive oil, corn oil or anhydrous milk fat prepared by blending dietary oils to a basal diet mix as previously reported, whereas the chow control contained 9% energy from corn oil [11]. Mice were raised in the same room and litter mates were separated into different diet groups post-natally and then co-housed with four mice per cage. From these four, two mice per cage were used in this study giving a total of 4 cages per group. Mice (Jackson Laboratories, Bar Harbor, Maine) were maintained at the Center for Disease Modeling at the University of British Columbia (UBC), Vancouver, Canada. The animal room was temperature controlled (22+/−2°C) with a 12-h light/dark cycle and fed with respective diets ad libitum with free access to autoclaved pH neutral water under a specific pathogen-free condition. Food intake and weight gain was monitored weekly. Mice were anaesthetized with isoflurane and euthanized by cervical dislocation. The distal region of the colon (with the luminal content and stool removed) was snap frozen in liquid nitrogen and stored at −80°C prior to amplicon sequencing/proteomic experiments.

### 2.2. Bacterial Genomic DNA Extraction

Frozen tissues were homogenized using stainless steel beads in Mixer Mill MM 400 (Retsch). Bacterial genomic DNA was extracted with QIAamp^®^ DNA Stool Mini Kit according to the manufacturer’s instructions. DNA concentration and purity were checked with Nanodrop 2000 (Thermo Scientific). Primers were used to amplify the 16S rRNA gene as described previously [19]. The PCR product was purified with QIAquick^®^ Gel Extraction Kit (Qiagen) and PCR amplicons concentration was normalized with SequalPrep^TM^ Normalization Plate Kit (Invitrogen). Library preparation, emPCR amplification and picotitre plate pyrosequencing using titanium chemistry was carried out by Vancouver Prostate Centre, UBC and Vancouver General Hospital Centre of Excellence in accordance with Roche/454 Life Sciences protocol on the 454 GS FLX+ System.

### 2.3. Bioinformatics Sequencing and Analysis

Sequencing was performed using Roche 454 technology. Sequences were analyzed using the Quantitative Insights Into Microbial Ecology (QIIME) pipeline [20] with default parameters. Since reads in the 454 platform vary in length the two runs (male and female colons) were truncated to a length of 250 to retain at least 70% of the reads with the recommended 1% expected error threshold [21]. Libraries were processed with a minimum quality score of 25 and a quality score window value of 50. The quality filtered reads were then combined and chimeras were filtered using usearch61 [22]. Sequences were aligned using PyNAST [23] and any sequences that failed to align were omitted from the subsequent tree and operational taxonomic unit (OTU) table. Both open-reference and closed-reference OTU clustering was done at 97% similarity level against the most recent GreenGenes database (gg_13_8_otus). An open-reference OTU table contains a combination of de novo OTUs (reads that do not match reference sequences) as well as reads that match sequences in the reference database. Closed-reference OTU table discards any reads that do not match the sequence in the reference database. Prior to sequence processing, the individual sequencing statistics for the male colon was 336,801 reads (max length 911, average 400.1) and 326,727 reads (max length 1190, average 421.9) for the female colon. Following quality filtering, truncation and chimera removal a mean total of 6586 high-quality bacterial 16S rRNA sequence reads from the 32 mice remained prior to rarefaction. Samples were rarefied to the same sequencing depth of 2069 (open-reference) for alpha diversity and beta diversity, and 1458 (closed-reference), for phyla ratios using QIIME2. Alpha diversity rarefaction curves were used to ensure appropriate sampling depth. Phylogenetic Investigation of Communities by Reconstruction of Unobserved States (PICRUSt) [24] and linear discriminant analysis effect size (LEfSe) [25] tools were used for further analyses.

### 2.4. Alpha and Beta Diversity Analysis

Alpha diversity metrics included observed species richness, Chao1, Simpson’s index (D) and Shannon’s diversity. A Kruskal–Wallis analysis combined with Benjamini–Hochberg adjustment for multiple comparisons was used to determine the gut microbiome differences between the dietary groups. The structure of bacterial communities in each diet group were compared using weighted and unweighted UniFrac metrics [26]. Based on these distance matrices, a PERMANOVA [27] was used to analyze sample composition. Significance was assessed by 999 permutations for all distance-based methods. An adjusted *P* value (Q-value) less than 0.05 was considered statistically significant. To visualize microbial community composition, a principal coordinates analysis (PCoA) was performed on the distance data and the first two principal components were used to generate an ordination plot in Primer 6.

### 2.5. Abundance Analysis

LEfSe was used to identify differences in taxa composition and Kyoto Encyclopedia of Genes and Genomes (KEGG) orthologs between the dietary groups. Differential abundance analysis was performed on the closed-reference OTU table with the logarithmic linear discriminant analysis (LDA) score of 2 as the cutoff and the less permissive ‘all-against-all’ strategy selected for pairwise comparisons [25]. LEfSe first tests for statistical significance between dietary groups (non-parametric Kruskal–Wallis test) followed by quantitative tests for biological consistency (non-parametric Wilcoxon-rank sum test). Multiple test corrections were performed by the Benjamini–Hochberg procedure-based false discovery rate (FDR) control (‘p.adjust’ in R). An adjusted *P* value (Q-value) less than 0.05 was statistically significant.

### 2.6. Amplicon Sequencing Prediction Analysis

PICRUSt was used to infer the relative abundance of gene families and biochemical pathways based on the 16S rRNA data (version: 13_5) [24] The rarefied closed-reference OTU table was first normalized for the 16S copy numbers of each OTU and then linked to KEGG annotations of reference genomes [28]. The generated KEGG pathways were submitted to HUMAnN (The HMP Unified Metabolic Analysis Network; version 0.99) for further analysis. The HUMAnN produced pathway summaries were analyzed by LEfSe to determine the differential abundance of KEGG pathways. BugBase [29], a microbiome analysis tool used to predict high-level phenotypes, was used to determine the proportion of Gram-positive, Gram-negative, aerobic, anaerobic, facultative anaerobic, biofilm forming and mobile element containing bacteria.

### 2.7. Short-Chain Fatty Acid Analysis

SCFAs were analyzed from cecal tissue samples using direct injection gas chromatography as previously described [19]. Tissue samples were homogenized in 700 μL of isopropyl alcohol, with 0.01% 2-ethylbutryic acid as the internal standard, at 30 Hz for 13 min using stainless steel beads. Homogenate was centrifuged at 15,100 × *g* for 10 min at 4 °C. Complete extraction was confirmed by absence of SCFA in the supernatant after second re-extraction of the remaining tissue pellet. 0.9 μL of cleared supernatant was directly injected to Trace 1300 Gas Chromatograph (D.I.A.B.E.T.E.S center, UBCO), equipped with flame-ionization detector, with AI1310 auto sampler in splitless mode. A fused silica FAMEWAX column 30 m × 0.32 mm i.d. coated with 0.25μm film thickness was used. Data analysis was carried out with Chromeleon 7 software. Peaks were analyzed on software and the area under peaks for acetic, propionic, and butyric acid data were represented as percent weight of total wet cecal sample (mass %).

### 2.8. Protein Extraction

Frozen colon pieces were scraped to separate the mucosa from the submucosa following a similar protocol as previously described [30]. The submucosal and the mucosal samples were separately put into lysis buffer made up of 25 mM HEPES solution (pH = 7.5) with 1 tablet protease inhibitor containing bestatin, AEBSF, EDTA, pepstatin, and E-64 (Thermo Fisher Scientific), 7 M urea, 2 M thiourea, and 4% CHAPS. The samples were homogenized via bead beating. Insoluble materials were removed by centrifugation and then soluble proteins were acetone precipitated from the supernatant and pelleted by centrifugation.

### 2.9. Protein Digestion, Itraq Labeling and LC-MS/MS Analysis

Samples were prepared for proteomic analysis at the University of Victoria, Genome BC Proteome Center located at the Vancouver Island Technology Park. Equal amounts of the extracted protein from each mouse were pooled, group-wise, to generate pooled lysates for low fat (n = 6), milk fat (n = 6), olive oil (n = 6) and corn oil (n = 6) groups. Further, equal amounts of protein from all dietary groups were used to generate a total protein’ pool. Sample pooling strategy has been used widely to reduce the effect of biological variation while dealing with clinical samples [31,32,33,34]. 100 µg of protein from each dietary group was trypsin digested and then individually labeled using 8-plex iTRAQ reagents (AB Sciex, ON, Canada). The labeled peptides were pooled and vacuum centrifuged until the final volume was approximately 100µL. An Agilent 1290 High-Performance Liquid Chromatography (HPLC) system (Agilent, CA, USA) was equipped with an XBridge C18 BEH300 (Waters, MA, USA) 250 mm X 4.6 mm, 5 µm, 300 A HPLC column. The flow rate was set to 0.75 mL/min, samples were injected onto the column and fractions were collected every minute for 96 min. The HPLC fractions were then reduced in volume by lyophilization and concatenated into 24 fractions by combining every 24th fraction. C18 StageTip concentrated samples were analyzed by reversed phase nanoflow HPLC with nano-electrospray ionization using a LTQ-Orbitrap Velos Pro mass spectrometer operated in positive ion mode with a 2 h reverse phase gradient per HPLC fraction. Each sample was rehydrated and samples were separated by on-line reversed phase liquid chromatography coupled on-line to an LTQ-Orbitrap Velos Pro mass spectrometer equipped with a Nanospray Flex source (Thermo Fisher Scientific). Spectrum Selection was used to generate peak lists of the higher-energy collisional dissociation (HCD) spectra (parameters: activation type: HCD; s/n cut-off: 1.5; total intensity threshold: 0; minimum peak count: 1; precursor mass: 350-5000 Da).

### 2.10. Protein Data Processing and Sequence Database Searching

All data was analyzed using Proteome Discoverer version 1.4. The peak lists were submitted to an in-house Mascot 2.4 (Matrix Science) server for database searching through the Proteome Discoverer software. All host data was searched against the mouse sequence database, Uniprot-Mouse database (43,908 sequences; 19,909,825 residues) using similar search parameters [35]. All bacterial data was searched against Bacteroidetes (11363 entries) and Firmicutes (17039 entries). Scaffold (version 4.6.1, Proteome Software Inc., Portland OR), a software suite from Proteome Software was used for statistical validation of MS/MS based peptide and protein identifications. Scaffold software provides different levels of blocking in proteome analysis. Blocking is a statistical tool used to reduce biases and minimize variances within a study. Scaffold provides four blocking levels, for example a single protein in the original observation matrix can be summarized in terms of all spectra, unique spectra, unique peptides and unique samples. Unique peptides is the preferred blocking level for analyzing the data [36], allowing users to compare measurements for each peptide. Since we pooled biological replicates to minimize biological variance, we do not have biological replicates in our study design. Therefore, we chose unique peptides as our statistical blocking method. Differential proteins, therefore, were predicted using the differential peptides determined by Scaffold. Peptide identifications were accepted if they could be established at greater than 95.0% probability by the Scaffold Local FDR algorithm and contained at least two identified peptides for the host proteome and at least one identified peptide for the bacterial proteome. Protein probabilities were assigned by the Protein Prophet algorithm [37]. Spectra data were log-transformed, pruned of those matched to multiple proteins and those missing a reference value, and weighted by an adaptive intensity weighting algorithm. Differentially expressed proteins were determined by applying Permutation Test with adjusted significance level *P* < 0.05 corrected by Benjamini–Hochberg.

### 2.11. Ingenuity Pathways Analysis for Mucosal Host Proteins

Ingenuity Pathways Analysis (IPA) was used to interpret the host proteome data in the context of biological processes, pathways and networks. IPA infers hypothetical protein interaction clusters using the Ingenuity Pathways Knowledge Base, a large database consisting of millions of individual relationships between proteins. Given its proximity to the microbiome, the host mucosal proteomics data derived from the iTRAQ experiment was converted by IPA to ‘fold change’ and then uploaded into the IPA program. No expression value cutoff was selected and both up- and down-regulated identifiers were defined as value parameters for the analysis. Heatmaps highlighting significant downstream biological processes that are increased or decreased based on gene expression results are displayed as canonical pathways. To further explore connections between dietary intake and expressed genes, hypothetical networks were generated followed by regulator effect analysis [38], using as many proteins from the input expression profile as possible. Other proteins from the database were used to fill out a protein cluster when needed for a highly connected network as previously published [39]. To identify mucosal phyla that correlate with the selected host proteins, a Spearman correlation matrix was generated and plotted as a heatmap.

### 2.12. Statistical Analysis

Data is presented as mean ± standard deviation unless otherwise stated. The data was tested for normality using Shapiro-Wilk test, and a Kruskal–Wallis non-parametric test with a Benjamini–Hochberg FDR-correction was used for comparing differences in the relative abundance of Gram-positive, Gram-negative, aerobic, anaerobic and facultatively anaerobic bacteria, Firmicutes to Bacteroidetes ratio, mobile elements and biofilm formers between dietary groups. SCFAs were assessed using a Kruskal–Wallis non-parametric test followed by a Dunn’s multiple comparison test.

### 2.13. Data Availability

16S rRNA gene amplicon sequencing data is made available in the Genbank (SRA study ID: SRP082836). Metaproteome data is made available via ProteomeXchange for submucosal data (PXD008165) and mucosal data (PXD008152).

### 2.14. Ethical Considerations

The protocols used were approved by the Animal Care Committee of UBC under the protocol A15-0240 and in direct accordance with guidelines drafted by the Canadian Council on the Use of Laboratory Animals.

## 3. Results

### 3.1. Dietary Lipid Type Affects Gut Microbial Diversity

Dysbiotic bacterial communities are often associated with low diversity [40], although a causal relationship has not been established. Since the microbial composition of feces and mucosal tissue have different microbiomes [41], we focused on the mucosal associated microbes as these microbes are most likely respond to the dietary changes and have been suggested to be a reservoir for keystone species that contribute to disease activity [42]. Both the milk fat and corn oil diets resulted in increased alpha diversity (Figure 1A). Specifically, observed species richness and Chao1 were increased with corn oil and milk fat exposure whereas the olive oil diet resulted in similar richness to the low-fat chow. Similarly, Shannon’s index revealed that milk fat had high richness and evenness compared to low-fat chow and olive oil groups. An increase in Simpson’s index, indicating a decrease in evenness, was observed in all high-fat diets compared to the low-fat chow. These patterns of alpha diversity aligned with comparisons between samples amongst dietary groups. The PCoA plot using the weighted UniFrac revealed three distinct clusters where the milk fat and corn oil groups clustered together and away from olive oil and the low-fat groups (Figure 1B). A total of 80.8% of the overall variation in taxon composition was attributed to dietary exposure, of which the first and second axes explained 71.6% and 9.2% of the total variation, respectively. While the permutational multivariate analysis of variance (PERMANOVA) based on the weighted UniFrac distance suggests that milk fat and corn oil groups are similar, the unweighted UniFrac showed separation between the milk fat and corn oil groups suggesting that while the dominant species in the groups are similar, the rare species in the milk fat and corn oil groups are unique from each other (Figure 1C). Overall, the various dietary lipids each uniquely predicted the microbial community composition that are present in the gut.

The differences in bacterial communities between the dietary cohorts were further evident when samples were ordered according to their Firmicutes to Bacteroidetes ratio. In agreement with previous literature [3,4], our results revealed that high-fat diets induce a microbiome with a high Firmicutes to Bacteroidetes ratio in the colon compared to the low-fat diet (Figure 2A), although we did not see a significant difference in body weight between any of the groups (Appendix A). These findings coincided with a predicted increase in the relative abundance of Gram-positive bacteria in the high-fat diets and a corresponding decrease in Gram-negative (Figure 2B). Additionally, our findings indicated there were changes to the relative abundances of facultative anaerobes, aerobes and anaerobes as a result of the type of fat feeding (Figure 2C). Olive oil diets associated with the least abundance of oxygen tolerating microbes, important given the hypothesis that oxygen tolerant microbes are abundant during gut stress [43]. Specifically, the predicted abundances of facultative anaerobic bacteria were higher in the low-fat dietary group compared to the olive oil and corn oil groups, and were higher in the milk fat group compared to the olive oil group. Finally, each dietary fat resulted in a unique set of taxa (Figure 2D). The low-fat diet had an increase in the abundance of Lachnospiraceae [Firmicutes (*P =* 0.01)], *Aldercretzia* spp. [Actinobacteria (*P* = 0.03)], family S24_7 [Bacteroidetes (*P* = 0.002)], and *Ruminococcus* spp. [Firmicutes (*P* = 0.005)]. Uniquely, olive oil resulted in an increased abundance of several Firmicutes including Clostridiaceae (*P =* 0.003), Peptostreptococcaceae (*P* = 0.01), Ruminococcaceae (*P* = 0.005), and Dorea spp (*P* = 0.003). In contrast, milk fat promoted different families of Firmicutes including Erysipelotrichales (*P* = 0.008) and several genera from *Ruminicoccus* (*P* = 0.003). Corn oil enhanced the abundance of Firmicutes family members from Turicibacteraceae (*P* = 0.008) in addition to *Coprococcus spp.* (*P* = 0.002). Overall, high-fat diets resulted in analogous modulation of the gut microbiota at higher taxonomic levels, but the type of fatty acid present in the dietary lipid uniquely altered the intestinal microbes at lower taxonomic levels.

### 3.2. Dietary Lipid Type Confers Core Functionality to Each Microbial Community

Since microbial compositions change according to type of lipid diets, we next investigated how lipids affect the functionality of the microbiota using amplicon sequencing predictions and comparing SCFA metabolites. Amplicon sequencing functional content was predicted from marker genes (16S rRNA) and LDA was performed (Figure 3). The low-fat chow was predicted to enrich functions of the microbiome that included lipopolysaccharide biosynthesis, vitamin and cofactor biosynthesis (including biotin metabolism, folate biosynthesis, pantothenate and CoA biosynthesis, lipoic acid metabolism, and riboflavin metabolism), protein export, and digestion and absorption. This suggests that all high-fat diets, despite the type of fatty acid, may have reduced capacity for vitamin biosynthesis and cofactor metabolism. The olive oil diet is predicted to result in a microbiome that have an increased potential for pyruvate metabolism, enhanced synthesis and degradation of ketone bodies, butanoate metabolism and propanoate metabolism, enhanced lipid metabolism and abundant RIG-I-like receptor signaling important for viral immune recognition. The corn oil diet is predicted to result in a microbiome with functions characterized by increased flagellar assembly, ABC transporters, lipid metabolism (glycerolipid metabolism, sphingolipid metabolism, linoleic acid metabolism), and carbohydrate metabolism for ATP production (pentose phosphate pathway, galactose metabolism, starch and sucrose metabolism, fructose and mannose metabolism). Two component systems are also predicted be increased in the corn oil diet, which controls cellular processes such as cell motility and virulence. This may suggest that corn oil may result in a microbiota that is more invasive. Predictions from the milk fat diets suggest the highest potential for bacterial chemotaxis and similar to the corn oil diet carbohydrate metabolism was also predicted to be higher. This included glycolysis and gluconeogenesis, glyoxylate and dicarboxylate metabolism, C5-branched dibasic acid metabolism, biosynthesis of unsaturated fatty acids and xenobiotic degradation (styrene, dioxin, and xylene). This suggests that milk fat results in a microbiota with increased capacity for energy harvest. Overall, the predicted functional analysis suggest that total calories from fat altered common functional characteristics of the microbiota and that the type of lipid uniquely affected additional characteristics.

To determine if virulence attributes of the gut microbiome were modulated by high-fat diets, we used 16S OTUs to categorize functionality. We found that all high-fat diets predicted an increase in the abundance of mobile genetic elements (Figure 4A); however, this finding should be interpreted cautiously as mobile elements are subject to microevolutionary processes and may vary over short periods of time [44]. The diets composed of corn oil, including the standard chow, predicted increased levels of biofilm formers (Figure 4B). Specifically, the abundance of biofilm formers in the low-fat diet was significantly higher than the abundance of biofilm formers in the milk fat and olive oil groups but were not statistically different from the corn oil dietary group. Since SCFA metabolism was predicted to be modulated based on the amplicon sequencing extrapolations, we examined the abundance of cecal acetic, propionic and butyric acid to understand if the predicted changes in metabolic pathways affected the bioavailability of SCFAs. We found that the milk fat group had similar levels of SCFA as the low-fat chow groups whereas both olive oil and corn oil groups resulted in a decreased abundance of acetic acid, important for lipid biosynthesis, and propionic acid, important for gluconeogenesis, compared to the low-fat chow, respectively (Figure 4C). A similar trend was observed with abundance of butyric acid. Overall, these results suggest that the microbiome’s ability to yield SCFAs resulting from high calories of fat can be compensated via exposure to milk fat.

### 3.3. Dietary Lipids Alter Microbial and Host Proteins in the Colon

To understand the interactions between lipid diets, gut bacteria and the host, we performed metaproteomics to examine microbial and host proteins associated with the colonic mucosa and submucosa. Over 300 bacterial proteins were identified in the mucosa (Appendix A) and 112 were identified in the submucosa based on single peptide hits (Appendix A). However, it is currently recommended that a minimum of four peptides are required to be matched for positive protein identification, to decrease the number of false positives [45]. Based on this recommendation the only bacterial protein we could positively identify was the mucosal molecular chaperone dnaK protein which is significantly upregulated in the corn oil group (0.4-fold increase) compared to the low fat and olive oil groups (Appendix A). In stark contrast to the bacterial proteome, 1956 host proteins were identified in the submucosa and 1749 were identified in the mucosa based on a single peptide hit. Of these, 676 and 390 were confidently identified using four or more peptides with a *P* value ≤0.006 in the submucosa (Appendix A) and mucosa (Appendix A), respectively. Overall, there was low homogeneity between the submucosal and mucosal proteins with only 127 proteins overlapping the two biological niches. Overlapping proteins (Appendix A) largely included proteins important for host fatty acid metabolism such as Apolipoprotein A-1, Apolipoprotein E, fatty acid-binding protein, fatty acid synthase and 2,-4-dienoyl-CoA reductase; proteins important for cellular function such as ribosomal proteins, anion exchange proteins and endoplasmic reticulum resident proteins; proteins involved in epithelial remodeling such as cadherin-17 and vinculin; and proteins involved in mucosal defense and immunity including complement C3, and mucin-2.

#### 3.3.1. High-Fat Diets Associated with Decreased Death Receptor Signaling and Apoptosis and tRNA Charging

Molecular crosstalk between the commensal microbiota and the intestinal epithelial cells occurs at the intestinal mucosal surface. As such, we focused our investigation on host proteins expressed in the mucosa. To understand higher ranking response pathways due to different lipid diets, mucosal proteins were evaluated using IPA [46] which identifies the most significant canonical pathways, biological functions, and networks. After generating the pathway comparison heat map, we ranked the effects of each diet and ordered the results in descending order based on the high-fat corn oil diet (Figure 5A). The IPA heatmap highlights that all high-fat diets have decreased predicted pathways associated with cell death, indicated by the down-regulation of apoptosis signaling and death receptor signaling pathways. These findings were based on the overall down-regulation of cell death proteins such as apoptotic chromatin condensation inducer 1, cytochrome c somatic, lamin A/C, spectrin alpha non-erythrocytic 1, calpain 1, mitogen-activated protein kinase 1 and heat shock protein family B (small) member 1 (Table 1). While not included in the IPA pathway, increased interleukin-1 receptor antagonist in the corn oil and milk fat group, has also been shown to reduce apoptosis. 

Transfer RNA charging was similarly down-regulated in all high-fat diets. This was based on the overall down-regulation of glutamyl-prolyl-tRNA synthetase, phenylalanyl-tRNA synthetase beta subunit, lysyl-tRNA synthetase, asparaginyl-tRNA synthetase, arginyl-tRNA synthetase, threonyl-tRNA synthetase, valyl-tRNA synthetase and tyrosyl-tRNA synthetase. In contrast, all high-fat diets, had upregulated peroxisome proliferator activated receptor (PPAR)α/ retinoid X receptor (RXR)α pathways compared to the low-fat control. IPA selected proteins used to determine PPAR activation included: acyl-CoA oxidase 1, apolipoprotein A1, cytochrome P450 family 2 subfamily C member 18, fatty acid synthase, glycerol-3-phosphate dehydrogenase 1, heat shock protein 90 beta family member 1, mitogen-activated protein kinase 1, and protein disulfide isomerase family A member 3. Overall, increased consumption of fat regardless of the saturation index, results in decreased cell death and tRNA charging signaling and increased PPARα/RXRα activation signaling.

#### 3.3.2. Corn Oil Diets Show Responses Indicative of Increased Energy Requirements and Oxidative Stress, and Decreased Barrier Function

Two of the most highly affected host pathways in the corn oil group were glycolysis I and oxidative phosphorylation, indicating increased energy demand in the mucosal epithelial cells of corn oil fed mice (Figure 5A). These pathways were determined through upregulated proteins involved in glycolysis, such as enolase 1, fructose-bisphosphatase 2 and triosephosphate isomerase 1 (Table 1). Similarly, proteins involved in oxidative phosphorylation such as ATP synthase F1 subunit beta, ATP synthase peripheral stalk subunit OSCP (oligomycin sensitivity conferral protein), cytochrome c oxidase subunit 5A, and nicotinamide adenine dinucleotide (NADH): ubiquinone oxidoreductase core subunit S3 were upregulated in the high-fat corn oil and milk fat dietary groups. Other pathways heightened in corn oil diets are nuclear factor (erythroid-derived 2)-like 2 (NRF2) mediated oxidative stress response, and glutathione-mediated detoxification. IPA determined NRF2 mediated oxidative stress in the n-6 PUFA rich diets through the expression of glutathione-disulfide reductase, glutathione S-transferase mu 3 and superoxide dismutase 1. In contrast to the low-fat diet, the high-fat diet also had increased expression of carbonyl reductase 1, ferritin heavy chain 1 and ferritin light chain. Carbonic anhydrase 3 (CA3) and aldehyde dehydrogenase (ALDH2) are similarly increased in the corn oil group but were not included in the NRF2-mediated oxidative stress pathway. Overall, increased consumption of n-6 PUFA diets show responses indicative of increased energy demands and oxidative stress. While all high-fat diets have down-regulated integrin-linked kinase (ILK) signaling, this was especially pronounced in the corn oil group which showed decreased expression of actinin alpha 1 and 4, desmoplakin, filamin A and C, fibronectin 1, mitogen-activated protein kinase 1, myosin heavy chain 9, 11, and 14, myosin light chain 9 and vinculin. This is important because ILK signaling has been found to be indispensable for barrier function [47]. Other proteins important for epithelial integrity include mucins and proteins involved in junctional complexes. Here, we found that in addition to the prior mentioned proteins, Mucin-2 (Muc2; fragments) and cingulin were down-regulated in the high-fat dietary groups, particularly in corn oil. This decreased barrier function is not limited to the epithelium. The pathways show decreased vascular endothelial growth factor (VEGF) signaling in the corn oil and olive oil group which corresponds with the network generated by IPA predicting that the corn oil diet would increase bleeding based on the down-regulation of cluster of calponin-1 (CCN1), filamin-a (FLNA), cluster of myosin-9 (MYH9), and cluster of plectin (PLEC) proteins and the upregulation of interleukin-1 receptor antagonist protein (IL1RN) and apolipoprotein E (Figure 5B). In contrast, low-fat diets had FLNA, MYH9, and PLEC and down-regulated IL1RN, CCN1 and APOE, and was predicted to inhibit bleeding pathways (Figure 5C). Significant mucosal networks generated by IPA also predicted that high-fat corn oil diets inhibit contractility of muscle based on the down-regulation of Sarcoplasmic/endoplasmic reticulum calcium ATPase (ATP2A2), Cluster of Creatine kinase M-type (CKM), Cluster of Desmin (DES), Cluster of Myosin-11 (MYH11), myosin-14 (MYH14) and vinculin proteins (VCL) (Figure 5D). Taken together, diets rich in corn oil appear to have decreased barrier function, increased oxidative stress and require increased energy for maintenance.

#### 3.3.3. Milk Fat Diet is Associated with Increased Inflammation and Compensating Restitution

The milk fat dietary group had upregulated acute phase response (Figure 5A). Acute phase proteins are defined as proteins which are increased by at least 25 percent during inflammation and includes proteins such as apolipoprotein A1, ferritin, haptoglobin, interleukin-1 receptor antagonist protein, and serpin family A member 3, which were all upregulated in the milk fat group (Table 1). While not included in the IPA derived pathway, alpha-1 acid glycoprotein 1 was similarly upregulated and is involved in acute phase response. The milk fat group was the only high-fat diet to have upregulated fatty acid β-oxidation 1 and sirtuin signaling. Previous studies have shown that the sirtuin signaling pathways link inflammation and metabolism particularly protective restitutive responses helping to resolve inflammation [48]. Of the 22 proteins utilized in the IPA sirtuin signaling pathways, 6 were upregulated in the milk fat group including ATP synthase F1 subunit beta, H1 histone family member 0, NADH: ubiquinone oxidoreductase subunit A9, NADH: ubiquinone oxidoreductase core subunit S3, and voltage dependent anion channel 1. In contrast to the corn oil and olive oil dietary groups, IPA did not generate a hypothesis to explain how activation or inactivation of regulators leads to an increase or decrease of function in the milk fat group. Given these data, milk fat appears to have increased expression of inflammatory pathways.

#### 3.3.4. Olive Oil Consumption Was Associated with Increased Cytoskeletal Dynamics

Similar to the low-fat group, the olive oil group had increased proteins involved in actin cytoskeleton signaling and epithelial integrity (Figure 5A) due to increases in actinin alpha 1, filamin A, fibronectin 1, myosin heavy chain 11, myosin light chain 9 and vinculin. Additionally, the olive oil group had upregulated cluster of protein Col6a3 involved in microfibril formation. Predicted gene interaction networks show that olive oil was associated with inflammation of liver (tumorigenesis of tissue), commonly caused by virial infections. This prediction was based on the down-regulation of peroxisomal acyl-coenzyme A oxidase 1 (ACOX1), apolipoprotein a-1 (APOA1), Sarcoplasmic/endoplasmic reticulum calcium ATPase 2 (ATP2A2), complement C3 (C3), hippocalcin-like protein 1 (HP), interleukin-1 receptor antagonist protein (IL1RN), microsomal triglyceride transfer protein large subunit (MTTP), and pyruvate carboxylase (PC) proteins (Figure 5E).

### 3.4. Microbial Taxa Associate with Host Proteins

To understand potential interactions between the bacteriome and host mucosal proteins, we evaluated Spearman correlations between mean phyla abundance and the selected mucosal proteins contributing to IPA pathways and networks (Appendix A). The heatmap shows that low expression of Proteobacteria and Tenericutes inversely correlates with apoptosis in high-fat diets. Increased relative abundance of TM7 in the corn oil group correlated with several host proteins involved in glycolysis, oxidative phosphorylation and NRF2-mediated oxidative stress response. With respect to the olive oil diet, higher relative abundances of Bacteroidetes positively correlated with 42% of the proteins involved in actin cytoskeleton signaling. Finally, 55% of the acute phase response proteins in the milk fat group were associated with the relative abundances of Proteobacteria, Tenericutes and Verrucomicrobia. However, because the peptides were pooled, there were only four observations available for the Spearman Rank correlation analysis, one for each diet group. As a result, we cannot realistically draw conclusions from this correlative data, but rather advocate for well-controlled and designed experiments that ask specific questions based on the observations made here.

## 4. Discussion

The mammalian gut has co-evolved with trillions of microorganisms, the collection of which is referred to as the gut microbiome. We have yet to understand how microbes succeed in the gut as a consortium and then co-exist in a community and affect the host responses. It has been hypothesized that several external factors, including diet, play a role in the host-microbe interaction in the gut. Several studies over the past few years have shown that a high-fat diet can lead to different gut microbial profiles, yet the effects on bacterial taxa and their functional responses caused by distinct types of fatty acids are not well understood. To define the specific changes in bacterial taxa as well as functional outputs, we analyzed the effect of commonly consumed dietary lipids on the colonic microbiome.

While there were differences between the high-fat and low-fat diets suggesting calories may play a role, these diets are not directly comparable since the macronutrients and micronutrients are different in the low-fat diet. By changing the amount of dietary fat, the proportion of carbohydrates and proteins automatically changes making it difficult to disentangle lipid driven changes. For example, the availability of dietary carbohydrates has been shown to modulate biofilm development [49] and acquisition of plasmids encoding relevant metabolic pathways (mobile genetic elements) [50] which could account for the predicted differences between the “low” and high-fat diets. Therefore, we focus on changes observed between the high-fat diets. We found that each type of dietary lipid distinctly affected the clustering effects of the microbial communities indicating that different taxa thrive with exposure to the types of fatty acids. While all high-fat diets caused an increase in the abundance of Firmicutes, each dietary lipid promoted specific taxa within the phyla with differing functions, indicating that different Firmicutes species thrive in the presence of different lipid substrates. Promoting the growth of certain bacterial species through diet, or prebiotics, has primarily been documented in carbohydrates [51], yet the potential for other macronutrients to act as prebiotics has largely been unexplored. Correspondingly, SCFA production was also altered as a result of the type of lipid consumed. Propionic acid and acetic acid were suppressed in the corn oil and olive oil group, respectively, whereas the milk fat group had similar levels of SCFA production as the low-fat control. Our data also indicates that higher species richness is observed in mice fed corn oil and milk fat diets; however, the differential composition and predicted functions of the gut microbiota do not seem to be associated with better health outcomes. This is apparent in the corn oil group which promoted a microbiota with high invasive and infection potential. In support of this, previous studies from our lab have shown that the corn oil diet promotes exacerbated immune-driven damage when challenged with *Citrobacter rodentium* [11], whereas olive oil consumption is protective [9] despite the low microbial diversity shown here. Therefore, diversity alone may not be a predictor for a better health.

The data presented in this work show that diet induced changes to the microbiome mirrors diet induced responses from the host. For instance, the predicted increase in invasive potential observed in the corn oil bacteriome parallels the predicted decrease in pathogen resistance and barrier function observed in the host. Specifically, the corn oil diet increased the microbial diversity in the gut that was predicted to increase microbial virulence traits such as increased microbial motility and bacterial signal transduction by two component regulatory systems. The host proteome indicated the protective barrier protein MUC2 was decreased alongside proteins important for tight and adherins junctions, and endothelial integrity (bleeding). Furthermore, decreased peristalsis (contractility of muscle), and increased oxidative stress response predicted in the corn oil group may be in response to increased microbial invaders. This supports the phenotype observed in previous studies using a similar diet which showed n-6 PUFA results in increased oxidative stress and tissue damage [11,52], increased inflammation and mortality during enteric infection [9,11], and metabolic insufficiencies [53]. Our data, in combination with previous literature, indicates that increased n-6 PUFA in the diet may be a risk factor for the development of a dysfunctional barrier in the gut. While descriptive, the data presented here provides a potential mechanism (bacterial-host interactions) by which corn oil, rich in n-6 PUFA, imparts toxicity in the gut. Indeed, an overabundance of dietary n-6 PUFA promotes chronic inflammation [54] and excessive consumption of n-6 PUFA is a risk factor for IBD in humans [10]. Prospective cohort studies conducted over a 5-year period demonstrated that PUFA positively associated with UC risk [55]. Similarly, retrospective case-control studies found increased levels of IBD in people consuming diets rich in n-6 PUFA [56]. Our research and others support the observations made here that n-6 PUFA tends to increase gut inflammation and damage resulting in an exacerbated colitis in several animal models [11,57,58,59,60,61]. Currently, we do not understand the mechanisms behind n-6 PUFA being detrimental during colitis but this study does reveal pathways that need to be investigated further.

Similar parallels between the microbiome and host responses were observed in the milk fat group. Specifically, the milk fat diet increased microbes in the gut whose functions are involved in carbohydrate and lipid metabolism. This was reflected in the host by increased proteins involved in fatty acid β-oxidation. Mounting evidence supports that sirtuins link metabolism and host inflammation. While inflammation is required to defend against invading organisms, compensatory mechanisms are required to prevent chronic inflammation. Host sirtuins, increased in the milk fat group, improve outcomes in chronic inflammatory diseases and sepsis by ‘mending’ the host or promoting restitution through immune repression and restoring homeostasis following stress responses [48]. Moreover, while the milk fat diet resulted in a host-microbe relationship that promoted host inflammation, there were no significant decreases in protective microbial SCFA responses suggesting that both the host and the commensal microbes promote a homeostatic inflammation-resolution cycle. This supports previous studies showing increased pathology in the milk fat group during infection but also increased compensatory protective responses, unlike the corn oil group [9].

This relationship between the microbiome, host and dietary lipids is not limited to the bacteriome and there is evidence that the virome may similarly be involved. In support of this, sequencing data predicted upregulated RIG-I-like receptor signaling pathways in the olive oil dietary group. The RIG-I-like family of pattern recognition receptors identify viral RNA [62], and are important for virus-host signaling crosstalk. Host mucosal proteins in the olive oil cohort also predicted liver inflammation (tumorigenesis of tissue) which is often caused by viral infections. Previous experiments have shown viral infections can inhibit C3 complement production [63], and that the loss of IL1RN can enhance susceptibility to viral infections [64], which may suggest an interaction between dietary olive oil, the host and the virome. However, 16S rRNA gene amplicon sequencing predictions face several limitations, one of which is the inability to study viral microbiome community members. As such, future studies should test this potential relationship under controlled settings while specifically targeting the virome.

## 5. Conclusions

Overall, we conclude that the type of dietary lipids distinctly impacts the gut microbiome. While high fat consumption has a distinct impact on the gut microbiota as compared to a normal chow diet, the type of fatty acids alters the relative microbial abundances where olive oil was most distinct from the corn oil and milk fat. The corn oil and milk fat diets shared similarities in diversity but had different functional characteristics. We show that the corn oil diet, rich in n-6 PUFA, resulted in a microbiome with enhanced predicted virulence and pathogenicity associated with increased host inflammation, oxidative stress and increased barrier dysfunction. While the milk fat diet, rich in SFA, resulted in a host-microbe relationship that promoted inflammation which could result in inflammatory induced intestinal damage, there was a compensatory protective response evident by the production of sirtuins and SCFAs. In marked contrast to both corn oil and milk fat, the olive oil diet, rich in MUFA, resulted in a host-microbe dynamic suggesting the involvement of the less-explored virome. These results suggest that fat type is an important consideration for gut health and not all high-fat diets are detrimental. However, given this study is descriptive in nature, future studies should focus on well-designed experiments unraveling the mechanisms of each lipid on gut health. These results have the potential to guide evidence-based nutrition recommendations for IBD patients who can suffer from nutrient deficiencies from overly restrictive dietary regimes including low-fat diets.

## Figures and Tables

**Figure 1 nutrients-11-00418-f001:**
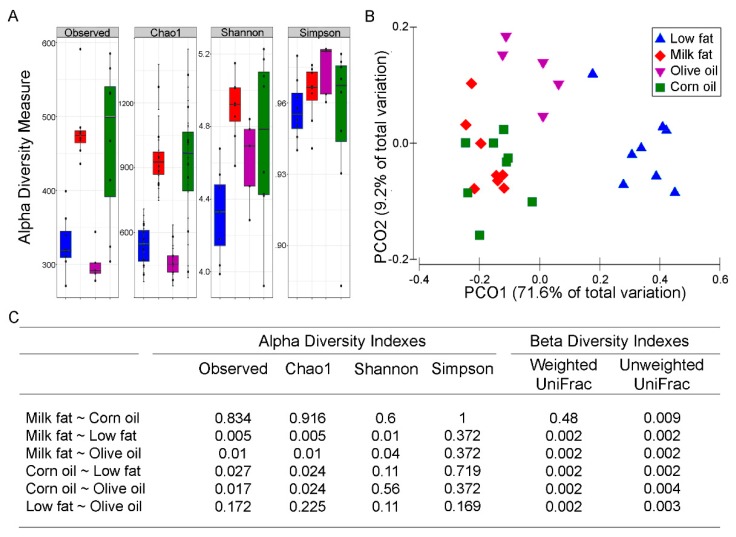
The effect of lipid diets on the diversity of the gut microbiota. (**A**) Alpha diversity of colonic microbiota from mice fed high fats diets composed of low-fat chow (blue), milk fat (red), olive oil (purple) or corn oil (green). Observed species richness, Chao1, Shannon’s, and Simpson’s indexes are displayed. (**B**) PCoA plot of the weighted UniFrac distances of colonic microbial communities from mice fed high-fat diets composed of milk fat, olive oil, corn oil or low-fat chow. The first two principal components from the PCoA are plotted. (**C**) Statistical summary (*p-*values after Benjamini–Hochberg adjustment for multiple comparisons) of all alpha and beta diversity measures.

**Figure 2 nutrients-11-00418-f002:**
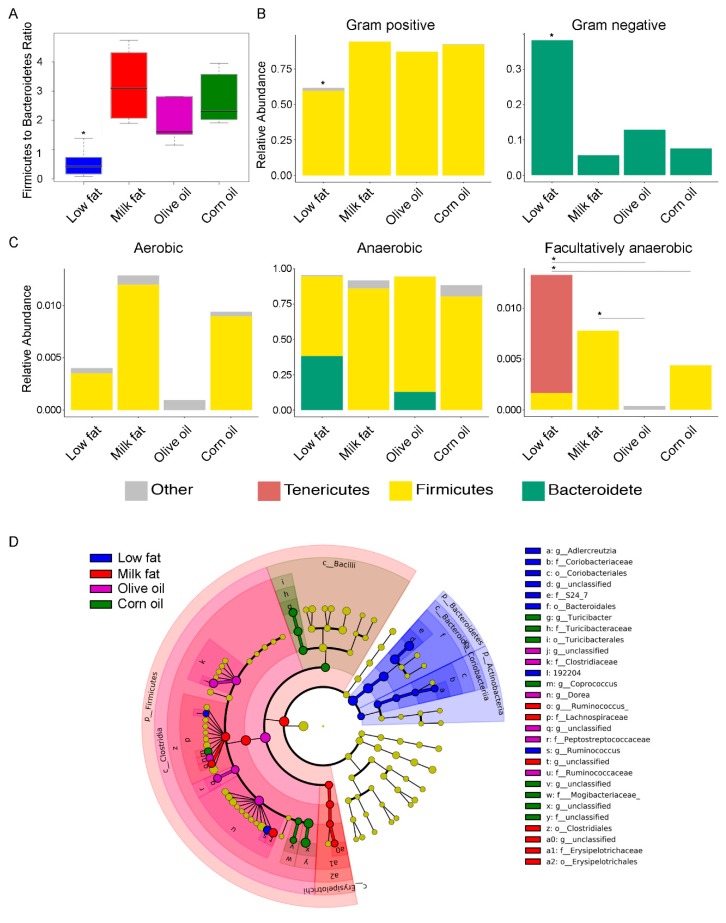
The effect of lipid diets on the gut microbial taxa. (**A**) Comparison of the log abundance of the Firmicutes to Bacteroidetes ratio among experimental diet groups in the colon. The y-axis of the box plot indicates the log of the abundance of the Firmicutes divided by the abundance of Bacteroidetes. The low-fat group had a significantly lower Firmicutes to Bacteroidetes ratio than all the high-fat diets. Within the high-fat diets, olive oil had a significantly lower ratio of Firmicutes to Bacteroidetes than the milk fat group. (**B**) Relative abundances of Gram-positive and Gram-negative bacteria in the diet groups show a significantly lower abundance of gram positive bacteria and a corresponding higher abundance of gram negative bacteria in the low-fat dietary group. (**C**) Relative abundances of aerobic, anaerobic and facultatively anaerobic bacteria in the diet groups show significantly lower facultative anaerobic bacteria in the olive oil group compared to the milk fat and low-fat group. An asterisk above a single column indicates *P* < 0.05 for that group compared to every other group. An asterisk with a line connecting two groups indicates *P* < 0.05 between the two groups. (**D**) Differentially abundant microbial clades in the colon microbiota from mice fed high-fat diets composed of anhydrous milk fat, olive oil, corn oil or a low-fat normal chow. Cladogram represents taxonomic representation of statistically and biologically consistent differences among lipid diet groups. Significant differences are represented in the color of the most abundant class. Yellow circles represent non-significant microbial clades. The all-to-all version of LEfSe was used with Kruskal–Wallis test (*P* < 0.05). LDA score threshold was set at default value 2.

**Figure 3 nutrients-11-00418-f003:**
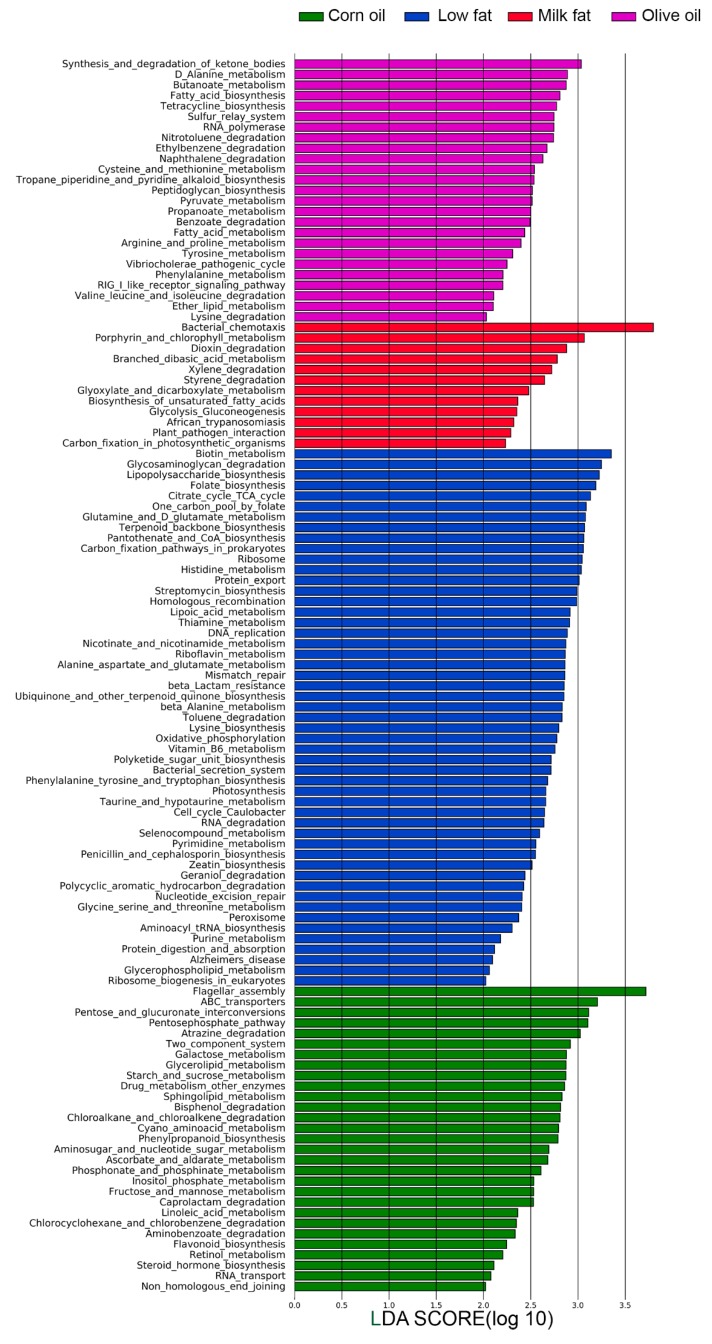
The effect of lipid diets on predicted microbial functions. Statistically and biologically differentially abundant pathways amongst the four dietary group shown as a histogram of the LDA scores. The length of the bars represents a log10 transformed LDA score set to a threshold value of 2. The one-to-all version of LEfSe was used with Kruskal–Wallis test (*P* < 0.05).

**Figure 4 nutrients-11-00418-f004:**
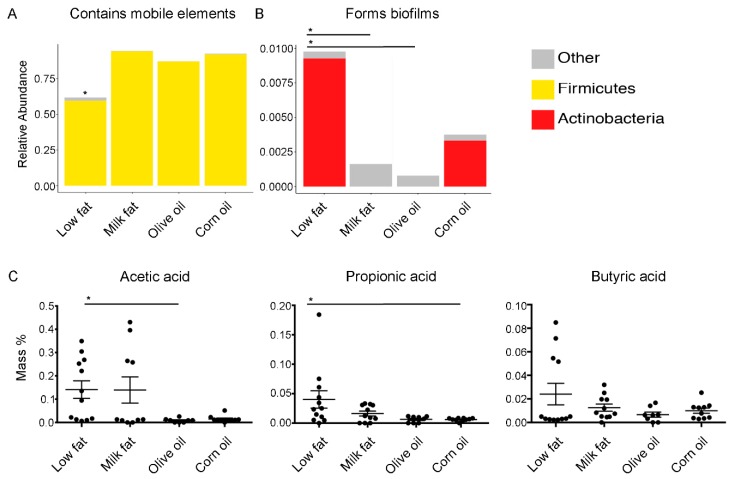
Predicted bacterial virulence traits and quantified secondary metabolites. Virulence traits such as (**A**) the relative abundance of bacteria which contain mobile elements and (**B**) the relative abundance of bacteria which are able to form biofilms are displayed for each diet group. (**C**) The effect of lipid diets on short-chain fatty acid production. Short-chain fatty acid analysis performed via gas chromatography on cecal samples from mice fed high-fat diets composed of milk fat, olive oil, corn oil or a low-fat chow. Acetic, propionic, and butyric acid are expressed as mass % of total cecal tissue sample. Values are expressed as mean +/− SEM (n = 8–12). An asterisk above a single column indicates *P* < 0.05 for that group compared to every other dietary group. An asterisk with a line connecting two groups indicates *P* < 0.05 between the two groups.

**Figure 5 nutrients-11-00418-f005:**
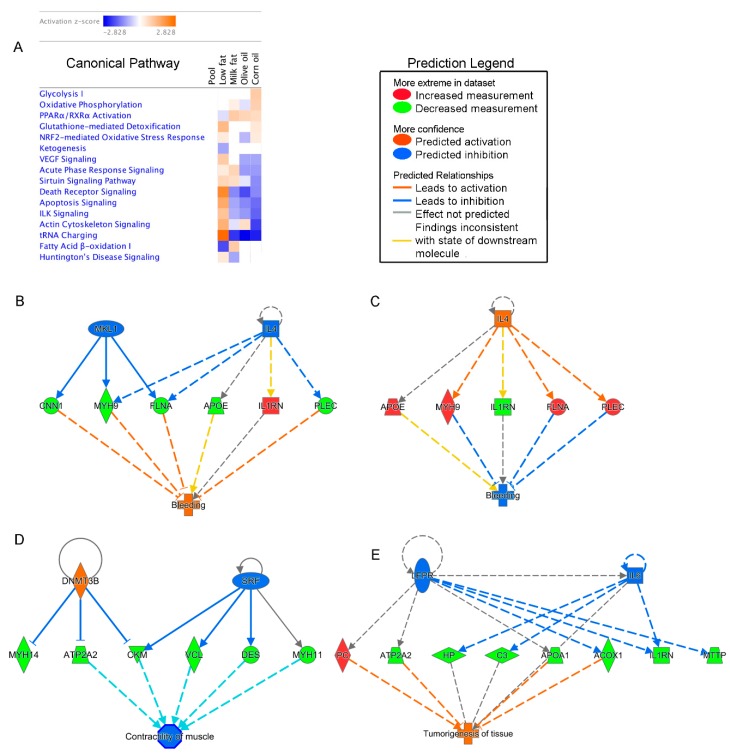
Effects of lipid diets on the gut proteome shown by the Ingenuity pathway comparative analysis. (**A**) Heatmap visualization of metabolites detected in each dietary group. Orange color indicates a higher activation score, whereas blue color indicates a lower activation score. Ingenuity pathway analysis (IPA) identified many upstream regulators predicted to be active based on the gene expression profile including: (**B**) bleeding in the high-fat corn oil and (**C**) low-fat chow groups, (**D**) contractility of muscles in the corn oil group and (**E**) tumorigenesis of tissue in the olive oil group. Faded colors represent less of an effect.

**Table 1 nutrients-11-00418-t001:** Mucosal proteins contributing to IPA pathways and networks.

Pathway	Symbol	Gene Name	Low Fat	Milk Fat	Olive Oil	Corn Oil
***High fat***						
Death Receptor	ACIN1	apoptotic chromatin condensation inducer 1	0.4	−0.2	0	−0.3
signaling	CYCS	cytochrome c, somatic	0.7	0.3	−0.2	−0.2
	HSPB1	heat shock protein family B (small) member 1	−0.5	0.5	0.1	−0.2
	LMNA	lamin A/C	0.6	−0.2	−0.1	−0.4
	SPTAN1	spectrin alpha, non-erythrocytic 1	0.6	−0.1	−0.1	−0.2
Apoptosis	ACIN1	apoptotic chromatin condensation inducer 1	0.4	−0.2	0	−0.3
	CAPN1	calpain 1	0.8	0	−0.1	−0.1
	CYCS	cytochrome c, somatic	0.7	0.3	−0.2	−0.2
	LMNA	lamin A/C	0.6	−0.2	−0.1	−0.4
	MAPK1	mitogen-activated protein kinase 1	0.2	0	−0.2	−0.1
	SPTAN1	spectrin alpha, non-erythrocytic 1	0.6	−0.1	−0.1	−0.2
	*IL1RN*	*Interleukin-1 receptor antagonist protein*	−*1*	*0.4*	*-0.1*	*0.7*
tRNA charging	EPRS	glutamyl-prolyl-tRNA synthetase	0.8	−0.1	−0.2	0
	FARSB	phenylalanyl-tRNA synthetase beta subunit	0.7	0	−0.2	−0.1
	KARS	lysyl-tRNA synthetase	0.8	−0.2	−0.3	−0.2
	NARS	asparaginyl-tRNA synthetase	0.9	−0.3	−0.5	−0.4
	RARS	arginyl-tRNA synthetase	0.7	−0.1	−0.2	0
	TARS	threonyl-tRNA synthetase	0.8	0	−0.2	−0.1
	VARS	valyl-tRNA synthetase	0.4	0	−0.1	−0.2
	YARS	tyrosyl-tRNA synthetase	0.5	−0.4	−0.3	−0.2
PPARa/RXRa	ACOX1	acyl-CoA oxidase 1	0.5	0	−0.3	−0.4
Activation	APOA1	apolipoprotein A1	−0.7	0.2	−0.1	0.5
	CYP2C18	cytochrome P450 family 2 subfamily C member 18	0.3	−0.4	0.4	−1.7
	FASN	fatty acid synthase	0	0	0	0.4
	GPD1	glycerol-3-phosphate dehydrogenase 1	1.3	−0.6	−0.4	−0.3
	HSP90B1	heat shock protein 90 beta family member 1	0.2	−0.4	−0.3	−0.1
	MAPK1	mitogen-activated protein kinase 1	0.2	0	−0.2	−0.1
	PDIA3	protein disulfide isomerase family A member 3	−0.5	0	0	0.1
***Corn oil***						
Glycolysis I	ALDOB	aldolase, fructose-bisphosphate B	1.4	−0.5	−0.6	−0.5
	ENO1	enolase 1	−0.5	0.2	0.1	0.3
	FBP2	fructose-bisphosphatase 2	0.2	−0.2	−0.2	0.1
	TPI1	triosephosphate isomerase 1	−0.6	0	0	0.4
Oxidative	ATP5F1B	ATP synthase F1 subunit beta	−0.8	0.2	0	0.5
phosphorylation	ATP5PB	ATP synthase peripheral stalk-membrane subunit b	0.6	−0.1	−0.1	−0.3
	ATP5PO	ATP synthase peripheral stalk subunit OSCP	−0.7	0.1	0.1	0.4
	COX5A	cytochrome c oxidase subunit 5A	−0.9	0.3	0.2	0.6
	CYCS	cytochrome c, somatic	0.7	0.3	−0.2	−0.2
	NDUFA9	NADH:ubiquinone oxidoreductase subunit A9	0.8	0.2	0.2	−0.1
	NDUFS1	NADH:ubiquinone oxidoreductase core subunit S1	−0.4	−0.2	0	0.4
	NDUFS2	NADH:ubiquinone oxidoreductase core subunit S2	0.6	0	0	−0.1
	NDUFS3	NADH:ubiquinone oxidoreductase core subunit S3	−0.8	0.1	0.1	0.3
	NDUFV2	NADH:ubiquinone oxidoreductase core subunit V2	−0.3	−0.1	−0.1	0.3
	UQCRB	ubiquinol-cytochrome c reductase binding protein	0.2	−0.3	−0.3	0
	UQCRC2	ubiquinol-cytochrome c reductase core protein 2	0.3	−0.1	−0.1	0.1
NRF2-mediated	CBR1	carbonyl reductase 1	−0.4	0.1	0.1	0.2
oxidative stress	CCT7	chaperonin containing TCP1 subunit 7	0.5	−0.2	−0.2	−0.3
response	DNAJB11	DnaJ heat shock protein family (Hsp40) member B11	0.7	−0.2	−0.4	−0.5
	FTH1	ferritin heavy chain 1	−0.4	0.3	0	0.1
	FTL	ferritin light chain	−0.2	0.2	0.2	0.3
	GSR	glutathione-disulfide reductase	0.6	−0.1	−0.2	0.1
	GSTM3	glutathione S-transferase mu 3	1.2	0.3	0.4	0.3
	MAPK1	mitogen-activated protein kinase 1	0.2	0	−0.2	−0.1
	SOD1	superoxide dismutase 1	0.6	−0.1	−0.2	0.2
	USP14	ubiquitin specific peptidase 14	0.6	−0.2	−0.2	−0.1
	*CA3*	*Carbonic anhydrase 3*	*0*	−*0.1*	*0*	*0.7*
	*ALDH2*	*Aldehyde dehydrogenase*	*−0.6*	*0.2*	*0*	*0.6*
Glutathione-mediated	ANPEP	alanyl aminopeptidase, membrane	1.8	−1.1	−0.8	−0.8
detoxification	GGH	gamma-glutamyl hydrolase	0.6	0.2	−0.1	0.6
	Gsta4	glutathione S-transferase, alpha 4	0.4	0	0	−0.5
	GSTM3	glutathione S-transferase mu 3	1.2	0.3	0.4	0.3
	GSTZ1	glutathione S-transferase zeta 1	−0.3	−0.1	0.1	0.5
ILK signaling	ACTN1	actinin alpha 1	0.3	0.1	0.1	−0.3
	ACTN4	actinin alpha 4	0.6	−0.3	−0.3	−0.2
	DSP	desmoplakin	0.6	−0.1	0	−0.3
	FLNA	filamin A	0.4	0.2	0.3	−0.4
	FLNC	filamin C	0.7	0.1	0.1	−0.6
	FN1	fibronectin 1	0.7	−0.1	0.1	−0.8
	MAPK1	mitogen-activated protein kinase 1	0.2	0	−0.2	−0.1
	MYH9	myosin heavy chain 9	0.6	−0.2	−0.2	−0.2
	MYH11	myosin heavy chain 11	0.7	0.2	0.4	−0.6
	MYH14	myosin heavy chain 14	0.5	−0.1	−0.1	−0.2
	MYL9	myosin light chain 9	−0.5	0.4	0.6	−0.2
	PPP2R1A	protein phosphatase 2 scaffold subunit Alpha	−0.5	0.1	0.2	0.5
	VCL	vinculin	0.6	−0.1	0.1	−0.4
Epithelial integrity	*Muc2*	*mucin-2*	*0.2*	*−0.1*	*−0.2*	*−0.6*
	*Cing*	*cingulin*	*0.6*	*−0.3*	*−0.2*	*−0.3*
VEGF signaling	ACTN1	actinin alpha 1	0.3	0.1	0.1	−0.3
	ACTN4	actinin alpha 4	0.6	−0.3	−0.3	−0.2
	EIF2S3	eukaryotic translation initiation factor 2 subunit γ	0.4	−0.1	−0.1	−0.2
	ELAVL1	ELAV like RNA binding protein 1	0.7	−0.1	−0.1	−0.1
	MAPK1	mitogen-activated protein kinase 1	0.2	0	−0.2	−0.1
	VCL	vinculin	0.6	−0.1	0.1	−0.4
Bleeding network	APOE	apolipoprotein E	−0.7	0.2	−0.1	0.5
	CNN1	cluster of calponin-1	−0.1	0.5	0.5	−0.2
	FLNA	filamin-a	0.4	0.2	0.3	−0.4
	MYH9	cluster of myosin-9	0.6	−0.2	−0.2	−0.2
	PLEC	cluster of plectin	0.4	−0.1	0	−0.5
	IL1RN	interleukin-1 receptor antagonist protein	−1	0.4	−0.1	0.7
Contractility of muscle network	ATP2A2	sarcoplasmic/endoplasmic reticulum calcium ATPase	0.5	−0.2	−0.1	−0.4
	CKM	cluster of creatine kinase M-type	−0.3	0.2	0.3	−0.3
	DES	cluster of desmin	0.6	0.4	0.3	−0.5
	MYH11	cluster of myosin-11	0.7	0.2	0.4	−0.6
	MYH14	myosin-14	0.5	−0.1	−0.1	−0.2
	VCL	vinculin	0.6	−0.1	0.1	−0.4
***Milk fat***						
Acute Phase Response	APOA1	apolipoprotein A1	−0.7	0.2	−0.1	0.5
	C3	complement C3	0.4	−0.1	−0.8	−0.7
	FN1	fibronectin 1	0.7	−0.1	0.1	−0.8
	FTL	ferritin light chain	−0.2	0.2	0.2	0.3
	HP	haptoglobin	0.8	0.2	−1.6	−1.3
	IL1RN	interleukin 1 receptor antagonist	−1	0.4	−0.1	0.7
	MAPK1	mitogen-activated protein kinase 1	0.2	0	−0.2	−0.1
	SERPINA3	serpin family A member 3	−0.6	0.6	−1.4	−1.1
	*AAG1*	*alpha-1 acid glycoprotein 1*	*0.8*	*0.4*	*-0.8*	*-0.6*
Sirtuin signaling	ADAM10	ADAM metallopeptidase domain 10	0.4	−0.1	−0.2	0
	APEX1	apurinic/apyrimidinic endodeoxyribonuclease 1	0.7	−0.1	−0.2	−0.3
	ATP5F1B	ATP synthase F1 subunit beta	−0.8	0.2	0	0.5
	ATP5PB	ATP synthase peripheral stalk-membrane subunit b	0.6	−0.1	−0.1	−0.3
	CPS1	carbamoyl-phosphate synthase 1	2.3	−2	−1.3	−1.9
	H1F0	H1 histone family member 0	−0.5	1.2	0.4	−0.6
	Hist1h1e	histone cluster 1, H1e	0.3	1.1	0.1	−0.6
	HMGCS2	3-hydroxy-3-methylglutaryl-CoA synthase 2	−0.4	0	−0.2	−1.5
	MAPK1	mitogen-activated protein kinase 1	0.2	0	−0.2	−0.1
	NAMPT	nicotinamide phosphoribosyltransferase	0.6	−0.1	−0.2	−0.3
	NDUFA9	NADH:ubiquinone oxidoreductase subunit A9	0.8	0.2	0.2	−0.1
	NDUFS1	NADH:ubiquinone oxidoreductase core subunit S1	−0.4	−0.2	0	0.4
	NDUFS2	NADH:ubiquinone oxidoreductase core subunit S2	0.6	0	0	−0.1
	NDUFS3	NADH:ubiquinone oxidoreductase core subunit S3	−0.8	0.1	0.1	0.3
	NDUFV2	NADH:ubiquinone oxidoreductase core subunit V2	−0.3	−0.1	−0.1	0.3
	PDHA1	pyruvate dehydrogenase E1 alpha 1 subunit	−0.3	0	0	0.1
	SF3A1	splicing factor 3a subunit 1	0.7	0	0.1	−0.1
	SLC25A5	solute carrier family 25 member 5	0.9	0	0	−0.2
	SOD1	superoxide dismutase 1	0.6	−0.1	−0.2	0.2
	TIMM13	translocase of inner mitochondrial membrane 13	−0.2	0	0	0.3
	UQCRC2	ubiquinol-cytochrome c reductase core protein 2	0.3	−0.1	−0.1	0.1
	VDAC1	voltage dependent anion channel 1	0.1	0.3	−0.1	0.2
Fatty acid B oxidation	ACAA2	acetyl-CoA acyltransferase 2	−0.4	0.1	0.2	0
	HADHA	hydroxyacyl-CoA dehydrogenase trifunctional multienzyme complex subunit alpha	−0.1	0.2	0.1	0.2
	HADHB	hydroxyacyl-CoA dehydrogenase trifunctional multienzyme complex subunit beta	−0.4	0.2	0.1	0.1
	IVD	isovaleryl-CoA dehydrogenase	−0.6	−0.1	0	0.2
***Olive oil***						
Actin cytoskeleton	ACTN1	actinin alpha 1	0.3	0.1	0.1	−0.3
signaling	ACTN4	actinin alpha 4	0.6	−0.3	−0.3	−0.2
	ARPC5	actin related protein 2/3 complex subunit 5	−0.4	0.2	0	0.4
	FLNA	filamin A	0.4	0.2	0.3	−0.4
	FN1	fibronectin 1	0.7	−0.1	0.1	−0.8
	IQGAP2	IQ motif containing GTPase activating protein 2	0.7	0	−0.1	−0.2
	MAPK1	mitogen-activated protein kinase 1	0.2	0	−0.2	−0.1
	MYH9	myosin heavy chain 9	0.6	−0.2	−0.2	−0.2
	MYH11	myosin heavy chain 11	0.7	0.2	0.4	−0.6
	MYH14	myosin heavy chain 14	0.5	−0.1	−0.1	−0.2
	MYL9	myosin light chain 9	−0.5	0.4	0.6	−0.2
	VCL	vinculin	0.6	−0.1	0.1	−0.4
	*Col6a3*	*cluster of protein Col6a3*	−*0.1*	*-0.6*	*1.1*	−*1.1*
Tumorigenesis of	ACOX1	acyl-coenzyme A oxidase 1	0.5	0	−0.3	−0.4
tissue network	APOA1	apolipoprotein a-1	−0.7	0.2	−0.1	0.5
	ATP2A2	sarcoplasmic/endoplasmic reticulum calcium ATPase2	0.5	−0.2	−0.1	−0.4
	C3	complement C3	0.4	−0.1	−0.8	−0.7
	HP	hippocalcin-like protein 1	0.8	−0.2	−0.2	0.2
	IL1RN	interleukin-1 receptor antagonist protein	−1	0.4	−0.1	0.7
	MTTP	microsomal triglyceride transfer protein large subunit	2.4	−2	−1.7	−2.1
	PC	pyruvate carboxylase	−0.2	0.2	0.1	0.4

Displayed are the experimental log ratios. Compared to the pooled reference channel (value normalized to zero), positive values indicate an increased fold-change expression whereas negative values indicate a decreased fold-change expression Italicized proteins were not included in IPA pathway. NA indicates the protein did not reach the 0.006 threshold. All groups are compared to the pool which was set to zero. Abbreviations used: OSCP, oligomycin sensitivity conferral protein; VEGF, vascular endothelial growth factor.

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
