# Peer review of "Gut Mucosal Proteins and Bacteriome Are Shaped by the Saturation Index of Dietary Lipids"

_nutrients, 2019, doi:10.3390/nu11020418_

Round 1
Reviewer 1 Report
This experimental research investigated the influence of intake of different types of lipids on gut microbiome, SCFA, and related proteomics. The research is of interest to the field.
The distal colon was biobanked for sequencing and proteomics study (Line 92). The SCFA was measured using cecal tissue. Please justify. Is microbial sequencing also done for mucosa or sub-mucosa compartment? The use of mucosa or sub-mucosa is not very clearly.
Did the investigators examine the potential gender difference?
Strength and limitation should be further discussed in one paragraph.
Shannon index, shannon’s index
Figure 1, A, what bar is for what dietary component, is it distinguished by color key in the same way as figure 1B.
Figure 1. C, Why the numbers do not match the scale on Y axis in Figure 1 A? It is not easy to understand.
What is the NC, AMF, OD, CD in Figure 2.
Line 270, need a “,”
What is the rationale to compare aerobic and anaerobic?
Table 1, the pathway is cut short. What does “-“, “+” mean? Is there any P value? The table is not easy to follow.
Line 544-545, not easy to follow.
Line 568? Augments? Lines 568-570, line 591.need rephrase.

Author Response
Comment 1. The distal colon was biobanked for sequencing and proteomics study (Line 92). The SCFA was measured using cecal tissue. Please justify. Is microbial sequencing also done for mucosa or sub-mucosa compartment? The use of mucosa or sub-mucosa is not very clearly.
Response 1. Fecal and cecal microbiotas are very similar (approximately 90% overlap in OTUs) and comparisons have shown that fecal microbiota can be effectively used to detect shifts and responses of cecal microbiota or vice versa (PMID: 25887695). Therefore, the SCFAs produced at each site are likely coming from the same microbes. The caecum and proximal colon are the main sites where fermentation is carried out and as a result, it is common to measure SCFAs from either the cecal content or fecal pellets. Logistically, we could not measure SCFAs from the colonic tissue because they were used up in the proteomics and sequencing experiments. We chose to use the cecal content over fecal pellets because SCFAs are rapidly absorbed in the large intestine (PMID: 30683) resulting in a lower concentration of SCFAs in feces. Any differences in SCFA concentrations in the fecal content could therefore be due to altered colonic absorption or colonic transit time. We believe that measuring SCFAs from the cecum reduces the impact of these factors and is more accurate than from fecal material.
The microbial sequencing was done on the whole tissue but would represent the mucosal microbiome. Mucus is a highly regenerative glycoprotein sheet secreted by host intestinal goblet cells. This mucus plays a key role in maintaining spatial host-microbial segregation whereby the inner mucus layer is relatively sterile and impenetrable. Therefore, in the absence of sepsis, the submucosa should not have microbes and the outer mucus layer forms the microbial niche in the gut.
Comment 2. Did the investigators examine the potential gender difference?
Response 2. We did explore sex effects in this study and had equal numbers of males and females. While we saw some trends to males and females responding differently to diet, we didn’t see any significant differences between males and females cautioning the sample size was too low (n=4 per sex) to draw any conclusions and therefore we elected to leave out the discussion. However, there is merit is looking at sex effects of diet and the microbiota in future studies.
Comment 3. Strength and limitation should be further discussed in one paragraph.
Response 3. As suggested, we have expanded the discussion of the limitations of this study (lines 627-629).
Comment 4. Shannon index, shannon’s index
Response 4. This has been made consistent throughout the paper.
Comment 5. Figure 1, A, what bar is for what dietary component, is it distinguished by color key in the same way as figure 1B.
Response 5. Yes the color key in Figure 1B corresponds to Figure 1A as well. We have clarified this in the figure caption.
Comment 6. Figure 1. C, Why the numbers do not match the scale on Y axis in Figure 1 A? It is not easy to understand.
Response 6. The values in Figure 1C are p-values after correcting for multiple testing. We have clarified this in the figure caption.
Comment 7. What is the NC, AMF, OD, CD in Figure 2.
Response 7. We have changed the figure legend to reflect the diets used in figure 2. We have uploaded a new figure 2.
Comment 8. Line 270, need a “,”
Response 8. Thank you, this has been corrected in the manuscript.
Comment 9. What is the rationale to compare aerobic and anaerobic?
Response 9. Normal adult microbiota thrives in the largely anaerobic environment, generating their own energy through fermentation.1 Olive oil diets associated with the least abundance of oxygen tolerating microbes which we point out is important given the hypothesis that oxygen tolerant microbes are abundant during gut stress [42]. This suggests a potential protective effect of olive oil and is mentioned on lines 294 of the Results.
Comment 10. Table 1, the pathway is cut short. What does “-“, “+” mean? Is there any P value? The table is not easy to follow.
Response 10. We thank the reviewer for their comment and have corrected the table to ensure the pathways are not cut short. Table 1 shows the experimental log ratios. Positive and negative values indicate that the protein expression was higher (positive) or lower (negative) than the pooled reference channel (normalized to zero). This has been clarified in the Table footnotes.
Comment 11. Line 544-545, not easy to follow.
Response 11. We have reworded this section for clarity.
Comment 12. Line 568? Augments? Lines 568-570, line 591.need rephrase.
Response 12. We have made the suggested changes.
Reviewer 2 Report
In this manuscript, Abulizi et al. feed mice with isocaloric diets containing different types of fat (milk fat, corn oil or olive oil) to investigate the influence of these different types of fat on the microbiota and on the host colonic proteome. They find that these distinct types of fat have differential effects on the microbiota composition, on the predicted metabolic functions of the microbiome and on the colonic host proteome. These analyses were well performed and the differential effects of the different types of fat are clear. This reviewer therefore agrees with the authors that not the calories but the types of lipids impact on host-microbe interactions in the gut.
However, from the microbiota alterations and the IPA network predictions of the host proteome alterations, the authors then conclude that among the different types of lipids it is specifically omega-6 PUFAs that underlie the effects they observe, and they conclude in the title of their manuscript that these lipids ‘predispose the colonic host-microbe interface to inflammation and damage’. Also in the abstract the authors mention far-reaching conclusions about omega-6 PUFAs increasing the potential for ‘pathobiont survival and invasion in an inflamed, oxidized and damaged gut’. This reviewer wonders where the authors found evidence for such claims. While corn oil indeed is rich in omega-6 PUFAs, there are many more differences in the composition and the metabolism of the different fat sources that could contribute to the effects observed in this study. The authors did not show any evidence for specific omega-6 PUFA effects in the mice. In addition, the IPA predictions are not sufficient to claim effects of the different fat types on complicated multi-factorial events such as ‘pathobiont survival and invasion in an inflamed, oxidized and damaged gut’. While well performed, nicely presented and clear, the data presented in this manuscript in my opinion are merely descriptive and do not justify drawing conclusions as the authors do. The conclusions of this manuscript are not sufficiently supported by the data provided.
Other comments:
· In Fig 3 and also in the discussion, the authors emphasise the effects of the olive oil diet on RIG-I signaling (not RIG-1, line 336). RIG-I signaling is a host innate immune response to viral infections, leading to IFN responses and inflammation. This reviewer does not understand how a host response can appear from 16S rRNA predictions of bacterial functionality. Can the authors explain this?
· In Fig 2D, for clarity reasons, it would be better to use the Low Fat, Milk Fat, Olive Oil and Corn Oil legends as in the other figures, instead of NC, AMF, OO can CO.
Author Response
Comment 1. However, from the microbiota alterations and the IPA network predictions of the host proteome alterations, the authors then conclude that among the different types of lipids it is specifically omega-6 PUFAs that underlie the effects they observe, and they conclude in the title of their manuscript that these lipids ‘predispose the colonic host-microbe interface to inflammation and damage’. Also in the abstract the authors mention far-reaching conclusions about omega-6 PUFAs increasing the potential for ‘pathobiont survival and invasion in an inflamed, oxidized and damaged gut’. This reviewer wonders where the authors found evidence for such claims. While corn oil indeed is rich in omega-6 PUFAs, there are many more differences in the composition and the metabolism of the different fat sources that could contribute to the effects observed in this study. The authors did not show any evidence for specific omega-6 PUFA effects in the mice. In addition, the IPA predictions are not sufficient to claim effects of the different fat types on complicated multi-factorial events such as ‘pathobiont survival and invasion in an inflamed, oxidized and damaged gut’. While well performed, nicely presented and clear, the data presented in this manuscript in my opinion are merely descriptive and do not justify drawing conclusions as the authors do. The conclusions of this manuscript are not sufficiently supported by the data provided.
Response 1. We thank the reviewer for their comment. Given these comments, we have changed the title to “Gut mucosal proteins and bacteriome are shaped by the saturation index of dietary lipids” to reflect that the types of lipid impact host-microbe interactions in the gut which we can confidently say from the data generated here. Our laboratory has a priori interest in omega-6 fatty acids. For decades the consumption of saturated fat was thought to increase cardiovascular risk and one of the proposed means to reduce cardiovascular disease was to reduce consumption of saturated fat. As a result, dietary intake of saturated fats has decreased and has been replaced with vegetable oils rich in omega- 6 PUFAs. However, a recent meta-analyses showed no association between saturated fat and cardiovascular disease risk,3,4 whereas previous publications from our lab using the same diets show omega-6 PUFA results in increased oxidative stress and tissue damage,5,6 increased inflammation and mortality during enteric infection, 7,8 and altered energy metabolism and fat oxidation.9 The data presented here provides a potential mechanism (bacterial-host interactions) by which omega-6 PUFAs impart their damaging effects previously reported in mice. In combination with previous literature, our paper indicates that increased n-6 PUFA in the diet may be a risk factor for the development of a leaky gut, which has clinical implications. From a pragmatic standpoint, IPA predicted more active upstream regulators in the corn oil group based on gene expression profiles, so we were more able to understand the predicted effect of corn oil on the gut proteome. With respect to the differences between the composition and metabolism of the different fat sources, we have attached the fatty acid analysis of the diets. As you can see, any of the differences in fats are in the medium and long-chain fatty acids reported (attached) so the only differences in the isocaloric and isonitrogenous diets we have used in the lipid source and in the corn oil the major difference is the level of n-6 PUFA; in the milk fat is the presence of SFA and in the olive oil is the abundance of MUFA. We do agree, that studying the individual fatty acid is not the same as studying the fat source so we were careful to discuss each group as the fat and not the fatty acid in the text.
Please see attachment for lipid profiles.
Comment 2. In Fig 3 and also in the discussion, the authors emphasise the effects of the olive oil diet on RIG-I signaling (not RIG-1, line 336). RIG-I signaling is a host innate immune response to viral infections, leading to IFN responses and inflammation. This reviewer does not understand how a host response can appear from 16S rRNA predictions of bacterial functionality. Can the authors explain this?
Response 2. Thank you for catching that typo, we have corrected this in the manuscript. PICRUSt uses evolutionary modeling to predict the functional capabilities of gene families in host-associated communities. While RIG-I signaling is typically reported as a host innate immune response to viral infections, some publications show that RIG-I detects mRNA of bacteria.10,11 The 16S rRNA predictions of RIG-I signaling have been reported previously.12
Comment 3. In Fig 2D, for clarity reasons, it would be better to use the Low Fat, Milk Fat, Olive Oil and Corn Oil legends as in the other figures, instead of NC, AMF, OO can CO.
Response 3. We have clarified this in the figure caption.
References:
1 Neish, A. S. Microbes in gastrointestinal health and disease. Gastroenterology 136, 65-80, doi:10.1053/j.gastro.2008.10.080
S0016-5085(08)01978-1 [pii] (2009).
2 Cabiscol, E., Tamarit, J. & Ros, J. Oxidative stress in bacteria and protein damage by reactive oxygen species. Int Microbiol 3, 3-8 (2000).
3 Chowdhury, R. et al. Association of dietary, circulating, and supplement fatty acids with coronary risk: a systematic review and meta-analysis. Ann Intern Med 160, 398-406, doi:10.7326/M13-1788
1846638 [pii] (2014).
4 Siri-Tarino, P. W., Sun, Q., Hu, F. B. & Krauss, R. M. Meta-analysis of prospective cohort studies evaluating the association of saturated fat with cardiovascular disease. Am J Clin Nutr 91, 535-546, doi:10.3945/ajcn.2009.27725
ajcn.2009.27725 [pii] (2010).
5 Ghosh, S., Molcan, E., DeCoffe, D., Dai, C. & Gibson, D. L. Diets rich in n-6 PUFA induce intestinal microbial dysbiosis in aged mice. The British journal of nutrition 110, 515-523, doi:10.1017/S0007114512005326 (2013).
6 Ghosh, S., Sulistyoningrum, D. C., Glier, M. B., Verchere, C. B. & Devlin, A. M. Altered glutathione homeostasis in heart augments cardiac lipotoxicity associated with diet-induced obesity in mice. J Biol Chem 286, 42483-42493, doi:10.1074/jbc.M111.304592
M111.304592 [pii] (2011).
7 Ghosh, S. et al. Fish oil attenuates omega-6 polyunsaturated fatty acid-induced dysbiosis and infectious colitis but impairs LPS dephosphorylation activity causing sepsis. PloS one 8, e55468, doi:10.1371/journal.pone.0055468 (2013).
8 DeCoffe, D. et al. Dietary Lipid Type, Rather Than Total Number of Calories, Alters Outcomes of Enteric Infection in Mice. The Journal of infectious diseases 213, 1846-1856. , doi:10.1093/infdis/jiw084 (2016).
9 Wong, C. K. et al. A high-fat diet rich in corn oil reduces spontaneous locomotor activity and induces insulin resistance in mice. J Nutr Biochem 26, 319-326, doi:10.1016/j.jnutbio.2014.11.004
S0955-2863(14)00247-2 [pii] (2015).
10 Schmolke, M. et al. RIG-I detects mRNA of intracellular Salmonella enterica serovar Typhimurium during bacterial infection. MBio 5, e01006-01014, doi:10.1128/mBio.01006-14
e01006-14 [pii]
mBio.01006-14 [pii] (2014).
11 Dixit, E. & Kagan, J. C. Intracellular pathogen detection by RIG-I-like receptors. Adv Immunol 117, 99-125, doi:10.1016/B978-0-12-410524-9.00004-9
B978-0-12-410524-9.00004-9 [pii] (2013).
12 Huang, Y. J. et al. Airway microbiome dynamics in exacerbations of chronic obstructive pulmonary disease. J Clin Microbiol 52, 2813-2823, doi:10.1128/JCM.00035-14
JCM.00035-14 [pii] (2014).

Reviewer 3 Report
General comment
Abulizi et al. described the impact of 3 different types of high fat diet on the diversity and abundance of the microbiota and on the colonic proteome of mice.
Although this is an important starting point for studying the impact of different types of dietary fat on the gut microbiota, data presented in this manuscript is too descriptive and speculative. Further discussion of these 2 points and a correlation analysis of the microbiota diversity with some gut proteome read-outs would mitigate these 2 points and strengthen the manuscript.
Specific comments
- Authors should clearly state the aim of the work.
- Authors used a diet where 40% of total calories came from fats. This is way above the % found in human diets (15-30%). Given this, how can authors discuss the implications of their findings in the context of nutritional recommendations for patients ?
- Why did authors did not check the cecal abundance of total Lactate?
- Lines 557-570: this part looks too speculative and should be re-written.
Author Response
Comment 1. Although this is an important starting point for studying the impact of different types of dietary fat on the gut microbiota, data presented in this manuscript is too descriptive and speculative. Further discussion of these 2 points and a correlation analysis of the microbiota diversity with some gut proteome read-outs would mitigate these 2 points and strengthen the manuscript.
Response 1. We thank the reviewer for their comment. We appreciate this study is indeed descriptive. However, we have made arguments based on the data presented here in the context of other studies reporting that n-6 PUFA is damaging to the gut. This is important, given that for decades the consumption of saturated fat was thought to increase cardiovascular risk and one of the proposed means to reduce cardiovascular disease was to reduce consumption of saturated fat. As a result, dietary intake of saturated fats has decreased and has been replaced with vegetable oils rich in omega-6 PUFAs. However, a recent meta-analyses showed no association between saturated fat and cardiovascular disease risk.3,4 Instead, an overabundance of dietary n-6 PUFA promotes chronic inflammation.8 Excessive consumption of n-6 PUFA is a risk factor for IBD in humans.10 Prospective cohort studies conducted over a 5-year period (n= 260,686) demonstrated that the only dietary factor positively associated with UC risk was PUFA consumption.11 Retrospective case-control studies found increased levels of IBD in people consuming diets rich in n-6 PUFA.12 Given this, our laboratory has a priori interest in omega-6 fatty acids since our research, as well as others, support the findings presented here where n-6 PUFA exacerbates murine colitis in several models.13-18 In addition to the direct effects of n-6 PUFA on gut inflammation, we have also shown that n-6 PUFA drives inflammation in part through its effect on gut microbes,13 and here our metagenomics/metaproteomic analysis revealed increased gene products that are known to stimulate gut inflammation. Previous publications from our lab using the same fat diets show diet rich in omega-6 PUFA results in increased oxidative stress and tissue damage,5,6 increased inflammation and mortality during enteric infection, 7,8 and altered energy metabolism and fat oxidation.9 The data presented here provides a potential mechanism (bacterial-host interactions) by which omega-6 PUFAs impart their damaging effects previously reported in mice. In combination with previous literature, our paper indicates that increased n-6 PUFA in the diet may be a risk factor for the development of dysfunctional barrier in the gut, which has clinical implications. Currently, we do not understand the mechanisms behind n-6 PUFA being detrimental during colitis but this study does reveal pathways that need to be studied further. We have added this to the discussion and elected to change the title to “Gut mucosal proteins and bacteriome are shaped by the saturation index of dietary lipids” to reflect that the types of lipid impact host-microbe interactions in the gut which can be confidently stated from the data shown here.
From a pragmatic standpoint, IPA predicted more active upstream regulators in the corn oil group based on gene expression profiles, so we were more able to understand the predicted effect of corn oil on the gut proteome.
In terms of correlation analyses, we agree that running a correlation matrix between phyla abundance and selected mucosal proteins would help migrate the two points and strengthen the manuscript. However, in order to make this comparison we would have to use the mean taxonomic data for each diet and compare it to the pooled protein data. This results in four data points and thus from a statistical standpoint, it is not feasible to draw strong conclusions from this data. While some interesting observations are made and discussed in the revised version of the manuscript, we elected to add the new figure as supplemental data shown in Figure S2. We advocate for well-controlled and designed experiments that ask specific questions based on the observations made from this analysis.
Figure S2 attached in the document.
Figure S2. Spearman rank correlation heatmap depicting correlations between selected host proteins and mucosal phyla. Colour indicates directionality (red = positive; blue = negative); cells outlined by a black box exceed the critical value (p < 0.05) using a two-tailed test.
Comment 2. Authors should clearly state the aim of the work.
Response 2. The aim of this work is to understand the effects of lipid diets on gut bacterial communities and their functional interaction with the host. This has been clarified in the text on lines 65-68.
Comment 3. Authors used a diet where 40% of total calories came from fats. This is way above the % found in human diets (15-30%). Given this, how can authors discuss the implications of their findings in the context of nutritional recommendations for patients ?
Response 3. Reported mean intake of fat in European countries range from 28.5-46.2% of total energy.13 Studies in France14, for example, routinely shows a 37-40% energy contribution from fat, making our diet relevant.
Comment 4. Why did authors did not check the cecal abundance of total Lactate?
Response 4. We only measured the SCFA that have well known roles in gut health.
Comment 5. Lines 557-570: this part looks too speculative and should be re-written.
Response 5. As suggested, we have re-written this portion.
References:
1 Neish, A. S. Microbes in gastrointestinal health and disease. Gastroenterology 136, 65-80, doi:10.1053/j.gastro.2008.10.080
S0016-5085(08)01978-1 [pii] (2009).
2 Cabiscol, E., Tamarit, J. & Ros, J. Oxidative stress in bacteria and protein damage by reactive oxygen species. Int Microbiol 3, 3-8 (2000).
3 Chowdhury, R. et al. Association of dietary, circulating, and supplement fatty acids with coronary risk: a systematic review and meta-analysis. Ann Intern Med 160, 398-406, doi:10.7326/M13-1788
1846638 [pii] (2014).
4 Siri-Tarino, P. W., Sun, Q., Hu, F. B. & Krauss, R. M. Meta-analysis of prospective cohort studies evaluating the association of saturated fat with cardiovascular disease. Am J Clin Nutr 91, 535-546, doi:10.3945/ajcn.2009.27725
ajcn.2009.27725 [pii] (2010).
5 Ghosh, S., Molcan, E., DeCoffe, D., Dai, C. & Gibson, D. L. Diets rich in n-6 PUFA induce intestinal microbial dysbiosis in aged mice. The British journal of nutrition 110, 515-523, doi:10.1017/S0007114512005326 (2013).
6 Ghosh, S., Sulistyoningrum, D. C., Glier, M. B., Verchere, C. B. & Devlin, A. M. Altered glutathione homeostasis in heart augments cardiac lipotoxicity associated with diet-induced obesity in mice. J Biol Chem 286, 42483-42493, doi:10.1074/jbc.M111.304592
M111.304592 [pii] (2011).
7 Ghosh, S. et al. Fish oil attenuates omega-6 polyunsaturated fatty acid-induced dysbiosis and infectious colitis but impairs LPS dephosphorylation activity causing sepsis. PloS one 8, e55468, doi:10.1371/journal.pone.0055468 (2013).
8 DeCoffe, D. et al. Dietary Lipid Type, Rather Than Total Number of Calories, Alters Outcomes of Enteric Infection in Mice. The Journal of infectious diseases 213, 1846-1856. , doi:10.1093/infdis/jiw084 (2016).
9 Wong, C. K. et al. A high-fat diet rich in corn oil reduces spontaneous locomotor activity and induces insulin resistance in mice. J Nutr Biochem 26, 319-326, doi:10.1016/j.jnutbio.2014.11.004
S0955-2863(14)00247-2 [pii] (2015).
10 Schmolke, M. et al. RIG-I detects mRNA of intracellular Salmonella enterica serovar Typhimurium during bacterial infection. MBio 5, e01006-01014, doi:10.1128/mBio.01006-14
e01006-14 [pii]
mBio.01006-14 [pii] (2014).
11 Dixit, E. & Kagan, J. C. Intracellular pathogen detection by RIG-I-like receptors. Adv Immunol 117, 99-125, doi:10.1016/B978-0-12-410524-9.00004-9
B978-0-12-410524-9.00004-9 [pii] (2013).
12 Huang, Y. J. et al. Airway microbiome dynamics in exacerbations of chronic obstructive pulmonary disease. J Clin Microbiol 52, 2813-2823, doi:10.1128/JCM.00035-14
JCM.00035-14 [pii] (2014).
13 Eilander, A., Harika, R. K. & Zock, P. L. Intake and sources of dietary fatty acids in Europe: Are current population intakes of fats aligned with dietary recommendations? Eur J Lipid Sci Technol 117, 1370-1377, doi:10.1002/ejlt.201400513
EJLT201400513 [pii] (2015).
14 Razanamahefa, L. et al. [Dietary fat consumption of the French population and quality of the data on the composition of the major food groups]. Bull Cancer 92, 647-657 (2005).

Round 2
Reviewer 2 Report
The authors have responded adequately to my questions. Changing the title of the manuscript improved the matching between the results and the conclusions. Although I still find some sentences in the abstract quite speculative when compared with the descriptive data, I now support publication of this manuscript. Congratulations to the authors!
Reviewer 3 Report
The authors adequately answered all my queries. The manuscript has been
improved and can be published in Nutrients.